# Diet Quality and Incident Non-Communicable Disease in the 1946–1951 Cohort of the Australian Longitudinal Study on Women’s Health

**DOI:** 10.3390/ijerph182111375

**Published:** 2021-10-29

**Authors:** Hlaing Hlaing-Hlaing, Xenia Dolja-Gore, Meredith Tavener, Erica L. James, Allison M. Hodge, Alexis J. Hure

**Affiliations:** 1School of Medicine and Public Health, University of Newcastle, Callaghan, Newcastle, NSW 2308, Australia; xenia.doljagore@newcastle.edu.au (X.D.-G.); meredith.tavener@newcastle.edu.au (M.T.); erica.james@newcastle.edu.au (E.L.J.); alexis.hure@newcastle.edu.au (A.J.H.); 2Hunter Medical Research Institute, New Lambton Heights, Newcastle, NSW 2305, Australia; 3Cancer Epidemiology Centre, Cancer Council Victoria, Melbourne, VIC 3004, Australia; allison.hodge@cancervic.org.au; 4Centre for Epidemiology and Biostatistics, Melbourne School of Population and Global Health, The University of Melbourne, Parkville, VIC 3010, Australia

**Keywords:** diet quality, non-communicable disease, multimorbidity, all-cause mortality, women

## Abstract

Diet quality indices (DQIs) can be useful predictors of diet–disease relationships, including non-communicable disease (NCD) multimorbidity. We aimed to investigate whether overall diet quality (DQ) predicted NCD, multimorbidity, and all-cause mortality. Women from the 1945–51 cohort of the Australia Longitudinal Study on Women’s Health (ALSWH) were included if they: responded to S3 in 2001 and at least one survey between 2004 (S4) and 2016 (S8), and had no NCD history and complete dietary data at S3. DQ was summarized by the Healthy Eating Index for Australian Adults-2013 (HEIFA-2013), Mediterranean Diet Score (MDS), and Alternative Healthy Eating Index-2010 (AHEI-2010). Outcomes included each NCD (diabetes mellitus (DM), coronary heart disease (CHD), hypertension (HT), asthma, cancer (except skin cancer), depression and/or anxiety) independently, multimorbidity, and all-cause mortality. Repeated multivariate logistic regressions were used to test associations between DQIs and NCD outcomes across the 15 years of follow-up. The mean (±sd) of DQIs of participants (*n* = 5350) were 57.15 ± 8.16 (HEIFA-2013); 4.35 ± 1.75 (MDS), and 56.01 ± 10.32 (AHEI-2010). Multivariate regressions indicated that women reporting the highest quintile of AHEI-2010 had lower odds of DM (42–56% (S5–S8)), HT (26% (S8)), asthma (35–37% (S7, S8)), and multimorbidity (30–35% (S7, S8)). The highest quintile of HEIFA-2013 and MDS had lower odds of HT (26–35% (S7, S8); 24–27% (S6–S8), respectively) and depression and/or anxiety (30% (S6): 30–34% (S7, S8)). Our findings support evidence that DQ is an important predictor of some NCDs and a target for prevention in middle-aged women.

## 1. Introduction

Globally in 2016, 71% of deaths were from non-communicable diseases (NCDs) [1]. More than one-third of those affected were aged 30–70 years, with diabetes mellitus (DM), cardiovascular disease (CVD), chronic respiratory disease (CRD), and cancer accounting for 4 out of every 5 deaths [1]. The presence of two or more chronic medical conditions or NCDs, known as multimorbidity, is more likely to be associated with premature deaths, limitations in physical functioning, and reduced quality of life than single conditions [2,3]. Generally, multimorbidity increases with age, especially among women [2], and is prevalent among middle-aged adults [4,5]. An Australian study estimated that 26% of people had NCD multimorbidity, among which DM, coronary heart disease (CHD), hypertension (HT), asthma, depression, and anxiety, were common [6].

Suboptimal diet is a risk factor for NCDs such as DM, CVD, and some cancers [7]. In the development of NCDs, obesity is another risk factor [8], and previous studies have shown that level of diet quality as measured by the Healthy Eating Index (HEI) is strongly and positively associated with obesity [8,9,10]. Furthermore, differential effects of socio-economic status (SES) on the relationship between diet and NCDs have also been reported [11,12]. Investigating the association between health outcomes and overall diet, rather than a single nutrient, is an important aspect of nutritional epidemiology [13,14]. Overall diet can be measured by constructing diet quality indices (DQIs) [13,15,16], in which a higher score typically represents better diet quality or adherence to dietary guidelines [17]. The earliest DQIs such as Diet Quality Index (DQI), Diet Quality Index-Revised (DQI-R), and HEI were based on the American Dietary Guidelines [18,19,20], whereas the Mediterranean Diet Score (MDS) was based on the Mediterranean dietary pattern [21]. Newer indices have been constructed based on country-specific dietary guidelines (e.g., Dietary Guideline Index (DGI) [22], China Dietary Guideline Index (CDGI) [23], or modified based on earlier indices (e.g., Recommended Food Score (RFS) [24], Australian Recommended Food Score (ARFS) [25], alternative Mediterranean diet (aMED) [26], etc.). Based on scientific evidence, the specific dietary pattern related to chronic diseases was measured as the Alternative Healthy Eating Index (AHEI) [24] and then regularly updated [27]. Both AHEI and the most updated Alternative Healthy Eating Index-2010 (AHEI-2010) have been used in epidemiological studies investigating the relationship between diet and health outcomes [28,29].

The associations between DQIs and various health outcomes have been extensively investigated [13,14,30,31,32,33,34,35,36,37]. The cumulative evidence suggests benefits of adherence to dietary guidelines (measured by Healthy Eating Index-HEI) and specific dietary patterns (measured by Alternative Healthy Eating Index-AHEI and Dietary Approach to Stop Hypertension-DASH) for DM, CVD, cancer, neurodegenerative diseases, and mortality [31]. Similarly, the Mediterranean diet, characterized by high intakes of vegetables, fruits, legumes, nuts, cereals, olive oil and fish, has been shown to be protective against DM [32], CHD [33], HT [34], cancer [35], depression [36,37] and all-cause mortality [38]. However, there is inadequate evidence regarding the association of overall diet and clustering or multimorbidity of NCDs, despite common risk factors or antecedents for many of the NCDs [39,40].

The aim of this project was to investigate whether diet quality (as measured using three different indices) predicted NCD (DM, CHD, HT, asthma, cancer (excluding skin cancer), depression and/or anxiety), including multimorbidity, and all-cause mortality, in women from the Australian Longitudinal Study on Women’s Health (ALSWH), born between 1946–1951. This study addressed the research question: Are the three DQIs (Healthy Eating Index for Australian Adult-2013 (HEIFA-2013), Mediterranean Diet Score (MDS), and Alternative Healthy Eating Index-2010 (AHEI-2010)) predictive of NCD outcomes? We hypothesised that women with high diet quality (as measured using HEIFA-2013, MDS and AHEI-2010) would have lower relative odds of NCDs (DM, CHD, HT, asthma, cancer (excluding skin cancer), depression and/or anxiety), multimorbidity, and all-cause mortality compared with those with low diet quality scores.

## 2. Materials and Methods

### 2.1. Study Population

Participants were from the Australian Longitudinal Study on Women’s Health (ALSWH), a national population-based study of more than 58,000 women across four birth cohorts: original cohorts included women born in 1973–1978, 1946–1951, 1921–1926 [41]; a new cohort of women born in 1989–1995 was subsequently added [42]. The three original cohorts of women were randomly sampled from the Medicare database, covering all citizens and permanent residents [41], with the more recent 1989–1995 cohort recruited using a combination of in-person, internet, and social media contact [42].

Ethical approval for ALSWH was granted by the Human Ethics committees of the University of Newcastle (approval number: h–076–0795) and University of Queensland (approval number: 200400224), and the study conforms to the ethical requirements of the Declaration of Helsinki. Further details about ALSWH are available on the study website (http://www.alswh.org.au) [43].

The current study used data from the 1946–51 cohort who completed baseline surveys in 1996 (survey 1, S1), a second survey in 1998 (survey 2, S2), and then every three years until 2016 (survey 8, S8). A total of 13,714 women aged 45–50 provided baseline data (S1, 1996), of which 8622 (63% of the S1 sample) completed S8 (2016). The participants were asked to provide information on their physical and mental health, health service uses as well as sociodemographic and lifestyle factors, including dietary intake information assessed by the Food Frequency Questionnaire (FFQ) known as the Dietary Questionnaire for Epidemiological Studies version 2 (DQES-v2) [44]. The DQES-v2 was included in S3 (in 2001) and DQI predictors were held constant across the study period.

In this study, data from S3 (2001, age: 50–55 years, *n* = 5350) were used as the baseline survey; however, sample selection was informed by S1 and S2, and covariates were included that were only measured at S1 (e.g., education). Women who responded to S3 in 2001 and at least one ALSWH survey between S4 (2004) and S8 (2016, aged 65–70 years) were included in the current study. Women were excluded if they had reported diagnosis of NCDs such as DM, CHD, HT, asthma, and cancer (except skin cancer) at or before baseline (S3). Women with missing FFQ data at S3 were also excluded. Figure 1 presents a simplified description of the study data for analysis. In total, 5350 women from S3 onwards were included for the final analysis (Figure 2). Further details on respondent numbers and loss to follow up from S4 to S8 are shown in Appendix A.

### 2.2. Dietary Assessment

The DQES-v2 is a semi-quantitative, validated FFQ [44] that was included in the ALSWH survey at S3 (2001). Nutrient intakes were estimated using data from the Australian NUTTAB95 nutrient composition tables [45]. Respondents were asked their usual intake over the last 12 months for 74 food items and six alcoholic beverages on a 1-to-10-point scale, ranging from “Never” to “≥3 times per day” for food items and “Never” to “Every day” for beverages. The validity of DQES-v2 was tested against a seven-day weighed food record and the resulting energy-adjusted correlation coefficients ranged between 0.28 and 0.70, demonstrating its adequacy for assessing habitual intake [46]. Items on the complete FFQ were converted to servings or grams per day for analyses.

### 2.3. Exposure Variables

The exposure variables were DQIs derived from the one DQES-v2 at S3, held constant across S4–S8. Baldwin et al. have shown diet quality to be relatively stable over the 12 years from S3 in 2001 to S7 in 2013 in the same ALSWH 1946–51 cohort, when the DQES-v2 was next repeated [47]. Therefore, in our analyses, diet at S3 was used to predict NCD outcomes in repeated cross-sectional analyses for S4 to S8.

Three DQIs were selected from our previous systematic review and critical appraisal [48]. The three indices reflect different theoretical approaches: (1) Healthy Eating Index for Australian Adults-2013 (HEIFA-2013) measures adherence to Australian Dietary Guidelines-2013 (ADG-2013) [49]; (2) Mediterranean Diet Score (MDS) measures adherence to a specific dietary pattern; and (3) Alternative Healthy Eating Index-2010 (AHEI-2010) measures consumption of foods and nutrients beneficial for chronic disease prevention. The computation and scoring system for each diet quality index (DQI) is provided in Appendix A.

The HEIFA-2013, the index based on ADG-2013 [49] was developed and validated in Australian young adults [50]. The HEIFA-2013 ranges from 0 to 100, with higher scores indicating greater adherence to the dietary guidelines. Full details of the validation and the scoring system are published elsewhere [50,51]. The HEIFA-2013 is composed of 11 components: 5 components of core food groups including grains (cereals), vegetables, fruits, milk and alternatives, and meat and protein alternatives; discretionary foods; 4 components of nutrients such as fats, added sugars, sodium, and alcohol; and water. For each component, the minimum score is zero and the maximum score is 10 points, except alcohol and water which contribute 5 points each. The scores for milk and alternatives, meat and protein alternatives, and discretionary foods were based on the number of servings consumed only. The scores for grains (cereals) were sub-divided into the number of servings of total grains (5 points) and the number of servings of whole grains (5 points). Moreover, the scores for vegetables and fruits were also subdivided into the number of servings consumed (5 points) and the variety (5 points). The scores were given incrementally based on ADG-2013 [49]: the maximum score was given for meeting the guidelines and the minimum score for not meeting the guidelines. For example, if a woman consumed <2.5 servings of discretionary food, she would receive 10 points; for 2.5–3.4 servings, she received 7.5 points; for 3.5–4.4 servings, she received 5 points; for 4.5–5.4, she received 2.5 points; and no points if she consumed ≥5.5 servings. Therefore, the minimum and maximum number of servings required for this study were: total grains (cereals) (1 to 6 servings), vegetables (1 to 5 servings), wholegrains (1 to 3 servings), fruit (0.5 to 2 servings), meat and protein alternatives (0.5 to 2.5 servings), milk and alternatives (0.5 to 2.5 servings), and discretionary foods (5.5 to 2.5 servings). The vegetable variety score was calculated if the respondents reported consumption of at least one serving (75 g) of green, orange, cruciferous, tuber, or bulb and 0.5 servings of legumes, with 1 point given for each variety. For fruit variety, 5 points were given for if 2 or more varieties of fruit were consumed. No points were given for scores below the minimum ranges.

For scoring of nutrients components, the fats component was scored for saturated fats (5 points) and mono-unsaturated fatty acid (MUFA) and poly-unsaturated fatty acid (PUFA) (5 points). Energy from saturated fat ≤ 10% scored 5 points, >10–12% scored 2.5 points, and >12% scored 0 points. Two servings of MUFA and PUFA scored 5 points; 0 servings scored 0 points. Sodium intake < 1610 mg scored 10 points; 1610 mg to 2300 mg scored 5 points; and >2300 mg scored 0 points. Although the updated food composition tables for Australia included “added sugar” [52], previously, there was no available data for “added sugar”. Therefore, percentage of energy from total sugar was used in this calculation and gave scores of 10 points to <5% total sugar; 5 points to 5–10% total sugar and 0 points to >10% of total sugar. Alcohol intake was scored as the number of standard drinks (10 g of alcohol) consumed and ≤2 standard drinks scored 5 points and >2 standard drinks scored 0 points.

Although water and other beverage intake was not assessed by the DQES-v2, the frequency of intake of cola, diet cola, other carbonated drinks, cordials, milk or soy milk, fruit or vegetable juices, tea, herbal tea, coffee, and water was included in S3. Water intake (including water/tea/herbal tea/coffee) was calculated as the proportion of water consumed in relation to other beverages with >50% scoring 5 points and 0% scoring 0 points, with each 10% increase between 0% and 50% scoring an additional 1 point.

The Mediterranean Diet Score (MDS), based on assessment of adherence to a Mediterranean diet, was developed and modified by Trichopoulou et al. [53]. The MDS can range from zero (minimum adherence) to nine (maximum adherence). This index was composed of nine components: vegetables, legumes, fruits, cereals, fish, lipid ratio, red meat and meat products, dairy products, and alcohol [53]. For calculation of the lipid ratio, both the sum of MUFA and PUFA were included in the numerator and saturated fat was included as a denominator. In this study, participants whose consumed amounts of vegetables, legumes, fruits, cereals, fish, and lipid ratio were below the sample median were assigned zero; those equal to or above scored one. Conversely, women who consumed red meat and meat products, and dairy products below the sample median were assigned one; those equal or above scored zero. Those who consumed alcohol in the range of 5 g to 25 g/day were assigned a value of one; above or below that range scored zero.

AHEI-2010 was based on foods and nutrients that lowered chronic diseases and modelled on Healthy Eating Index (HEI) [27]. The AHEI-2010 scoring ranges from 0 to 110, with a higher value representing healthier eating habits. Full details of the scoring system are published elsewhere [27]. It is composed of 11 components: vegetables, fruit, whole grains, nuts and legumes, long chain omega-3 fats, and PUFA considered as positive; intakes of sugar-sweetened beverages (SSBs) and fruit juice, red and processed meat, trans- fat, and sodium considered as negative; and alcohol intake considered as part of healthy diet if consumed in moderation. Each component was given a score between zero (less healthy diet) and 10 (healthier diet), with intermediate values scored proportionally to their intake. The minimum and maximum scoring for positive components were: vegetables (0–5 servings/day), fruit (0–4 servings/day), whole grain (75 g/day), nuts and legumes (0–1 serving/day), long chain omega-3 fats (0–250 mg/day) and PUFA (2–10% of energy). Conversely, participants who consumed SSBs and fruit juice (≥1 serving/day), or red and processed meat (≥1.5 servings/day), or trans-fat (≥4% of energy) or sodium (highest decile (mg/day)) were given minimum score. Those consuming no SSBs and fruit juice, or no red and processed meat, or ≤0.5% of energy from trans-fat or lowest decile (mg/day) of sodium were given maximum score. For alcohol intake, women consumed ≥2.5 drinks/day were assigned the minimum score and 0.5–1.5 drinks/day were assigned the maximum score.

### 2.4. Outcome Variables

The main outcome variables were:The incidence of NCDs (DM, CHD, HT, asthma, cancer (excluding skin cancer), depression and/or anxiety; incident cases following S3, with cases accumulating over time);Multimorbidity (defined as the co-existence of two or more of the above NCDs);All-cause mortality (new deaths since the last survey).

The occurrence of common NCDs was self-reported by participants and has been shown to be reliably reported against administrative data [54]. At S1, women were asked whether they had been diagnosed with either diabetes (high blood sugar), heart disease, or hypertension (high blood pressure), or asthma or cancer (breast, cervical, lung, bowel, and skin cancer) by using the question “Have you ever been told by a doctor that you have? (Circle one number on each line)?” At S2, their disease status of non-insulin dependent (type 2) diabetes, heart disease, hypertension (high blood pressure), asthma, cancer (breast, cervical, bowel, skin, and other cancer), depression, and anxiety were assessed by asking “Have you ever been told by a doctor that you have…? (Mark as many as applicable. Leave blank if you have never had this problem).” The responses included “yes, in the last 2 years” and “yes, more than two years ago”. From S3 to S8, they were asked whether they had been diagnosed with common NCDs since the last survey (i.e., in the last 3 years) by using the question “In the past three years, have you been diagnosed or treated for: (Mark all that apply)”.

NCDs were treated as reported by the women and also as enduring conditions [55], meaning that once a diagnosis was reported it was retained regardless of subsequent survey reports, except for depression and/or anxiety (Appendix A). Incidence of multimorbidity was defined as the co-existence of two or more of the selected NCDs (i.e., any combination of DM, CHD, HT, asthma, cancer (excluding skin cancer), and depression and/or anxiety) from S4 in 2004 to S8 in 2016. For the main analysis, women who had not reported DM, CHD, HT, asthma, and cancer (excluding skin cancer) at or before S3 in 2001 were included, to ensure the measure of diet quality preceded the health outcomes of interest, creating a temporal sequence.

All-cause mortality was assessed from the ALSWH participant status and cause of death data that had been linked with the Australian National Death Index. Any cause recorded on the death certificate was counted. Incident deaths were new deaths recorded between surveys; however, total deaths across S4–S8 were also modelled.

### 2.5. Covariates

Directed Acyclic Graphs (DAGs) [56,57] were constructed to illustrate the assumptions regarding causal relationship between exposure (diet), outcome (NCDs), and covariates across surveys (S4 to S8, 15 years of follow-up), and to identify confounders and mediators for statistical modelling. Based on the DAGs, the unbiased estimate of the relationship between DQIs and NCD multimorbidity, and all-cause mortality could be obtained. The DAGs show the main exposure (diet quality, measured once at S3) and the main outcomes (NCD, multimorbidity and all-cause mortality at S4–S8). Socioeconomic status (residence status, marital status, education, occupation, ability to manage income) and lifestyle variables (smoking, physical activity, taking prescribed and over-the-counter medicine) affect both exposure and outcome and hence are considered as confounders. Diet quality can influence body mass index, which can influence the risk of NCD multimorbidity and all-cause mortality; therefore, it is considered as the mediator and not adjusted for in models. The schematic illustration showing the association between exposure and outcomes, and DAGs are included in Appendix A.

The following time-varying covariates during the 15-year follow-up (S4 to S8) were included: age, area of residence (urban, inner regional, outer regional/rural), marital status (not married, married/de facto, separated/divorced/widow), education (no formal education, certificate (intermediate/high school), certificate (apprenticeship/diploma), university/higher degree), occupation (no paid job, paid job), ability to manage income (easy/not bad, difficult), smoking status (never smoked, history of smoking, currently smoke), physical activity (nil/sedentary (0–39 MET min/week), low (40–599 MET min/week), moderate (600–1199 MET min/week), high (≥1200 MET min/week)) [58], and taking prescribed medicine (no, yes), taking over-the-counter (OTC) medicine (no, yes). Since education status was assessed at S1 and S6, rather than every survey, participants’ education status at S3, S4 and S5 were created by using the value at S1 and S6; at S7 and S8 were created by using the value at S6. Missing items for covariates (range: 0.1–9.9%) (area of residence, marital status, education, occupation, ability to manage income, smoking status, physical activity, taking prescribed medicine, taking OTC medicine) were replaced applying the carry forward approach from the subsequent survey (for S4 to S7) and the response from the preceding survey (for S8).

Use of prescribed and OTC medicines was assessed inconsistently across the ALSWH surveys. Use of prescribed medicine during the past 4 weeks was assessed as number of different types of medicine (S1, S2), binary response (S5 to S8), and multiple responses (S3, S4). Likewise, women’s OTC medicine use status during the past 4 weeks was assessed as number of different types of medicine (S1 to S4), binary response (S5 to S8), and multiple responses (S3, S4). To harmonise these approaches, we generated binary variables for use of both prescribed and OTC medicine. Detailed information about ALSWH survey variables can be found at http://www.alswh.org.au/for-researchers/data [59].

### 2.6. Statistical Analysis

Statistical analyses were performed using Stata version 15. We performed repeated cross-sectional analyses using the DQI measures from S3 (held constant across subsequent surveys) to predict outcomes at S4 to S8, with adjustment for covariates from S4 to S8. Descriptive statistics of the baseline sociodemographic and lifestyle characteristics of women (at S3) were expressed as mean± standard deviation (sd) or *n* (%). Association between sociodemographic, lifestyle variables (at S3), and DQIs (at S3) were tested using analysis of variance (ANOVA) for continuous variables and chi-squared test for categorical variables. All-cause mortality and multimorbidity of NCDs of interest (DM, CHD, HT, asthma, cancer (excluding skin cancer), depression and/or anxiety) was determined. DQIs were calculated and described as a continuous measure (mean ± sd) and categorical measure (quintiles). Baseline characteristics of participants (at S1) that were included and excluded from the current study were also compared to explore potential selection bias (Appendix A).

Assuming the participants’ DQ were consistent across the 15 years of follow-up (five surveys, S4 at 2004 to S8 at 2016), association between DQIs at S3 and NCD outcomes were tested using bivariate (unadjusted) models. Multivariate models (adjusted for covariates at five surveys; sociodemographic variables such as age, residence, marital status, education, occupation, ability to manage income, and lifestyle variables including smoking, physical activity, and taking prescribed and OTC medicine) were fitted for effect estimates of DQIs on NCD outcomes (each disease and multimorbidity). The odds ratios and 95% CI for NCD outcomes (each disease and multimorbidity) with respect to DQIs were calculated considering the lowest quintile as the reference category. For the all-cause mortality outcome where numbers were low, quintile 4 and 5 of the DQIs were collapsed into one category, indicating high diet quality. The association between DQIs and all-cause mortality was reported as the ratio of the highest category (quintile 4 + 5) to the lowest category (quintile 1). Multivariate logistic regression models [60] were used to investigate the association between diet quality and NCD outcomes (each disease and multimorbidity) and all-cause mortality.

To check the robustness of the findings, sensitivity analyses using a subset of cases were undertaken, including *n* = 4026 women who remained at S8, 2016 for NCDs (each disease and multimorbidity), and *n* = 3032 women who ever had any NCD from S3, 2001 to S8, 2016 for all-cause mortality (Appendix A). The association between DQIs and total deaths accumulated across S4–S8 were investigated (Appendix A). A *p* value < 0.05 is considered significant, with consideration given to multiple comparisons, and all statistical tests are two-sided.

## 3. Results

The current study included 5350 women who were free of NCDs at S3 in 2001. The mean (±sd) of DQIs for the sample at S3 were 57.15 ± 8.16 (range: 25.42–86.67) for HEIFA-2013; 4.35 ± 1.75 (range: 0–9) for MDS, and 56.01 ± 10.32 (range: 26–93.75) for AHEI-2010, respectively. Across three DQIs, women with higher scores had university/higher degree, the ability to manage income easily, never smoked, and were more physically active than women with lower scores. Compared with their counterparts with DQI quintile 1, those in quintile 5 of HEIFA-2013 and AHEI-2010 were older and more likely to be in paid work; those in quintile 5 of MDS and AHEI-2010 had a healthy weight range and were more likely to have taken over-the-counter medicine. Those in the highest AHEI-2010 quintile were more often from an urban area (Table 1).

Table 2 shows results from univariate and multivariate logistic regression models estimating the association between DQIs and risk of common NCDs (including multimorbidity). Compared with the lowest quintile of DQIs, women reporting the highest quintile had reduced odds of NCDs (each disease and multimorbidity), especially in the later surveys. From S5 to S8 (2007 to 2016), those with AHEI quintile 5 had 42–56% reduced odds of DM comparing those with quintile 1. Likewise, we found a long-term protective effect of DQIs on HT (i.e., decreased odds of HT among those with higher scores of three DQIs; HEIFA-2013 (S7: 26%, S8: 35%), MDS (S6: 27%, S7: 25%, and S8: 24%, respectively) and AHEI-2010 (S8: 26%)) (Table 2).

The reduced odds of having asthma at S7 and S8 were found among those in the highest quintile of AHEI-2010 compared with those in the lowest quintile. Those in the highest quintile of HEIFA-2013 and MDS had reduced odds of depression and/or anxiety, HEIFA-2013 (at S6 30%, 95% CI: 0.51–0.98); MDS (at S7 30%, 0.49–0.96; and at S8 34%, 0.46–0.94), respectively. Lower odds of having NCD multimorbidity at S7 and S8 were found among those with higher scores of HEIFA-2013 and AHEI-2010 (Table 2).

Although we did not find any association between DQIs and CHD in multivariate models, there were univariate inverse associations (at S5: MDS; at S6: HEIFA-2013, MDS and AHEI-2010; at S7 and S8: AHEI-2010). However, we did not find any statistically significant associations for occurrence of cancer (excludes skin cancer) with the three DQIs (Table 2).

Table 3 shows results from univariate and multivariate logistic regression models estimating the associations between the three DQIs and all-cause mortality. Participants in the highest category of AHEI-2010 (quintile 4 + 5) compared with the lowest category (quintile 1) had lower odds of all-cause mortality at S4 for all women, in univariate analysis.

Comparing participants with the highest and lowest DQIs category, the odds of having NCDs (each disease and multimorbidity) among those who remained in the study at S8 and the odds of all-cause mortality among those who ever had any NCD during the study period (S3 to S8) were also consistent (Appendix A).

## 4. Discussion

Within this sample of middle-aged women, statistically significant associations between DQIs and NCDs, multimorbidity, and all-cause mortality were observed: DM (AHEI-2010 only); HT (all three DQIs); asthma (AHEI-2010 only); depression and/or anxiety (HEIFA-2013 and MDS); multimorbidity (HEIFA-2013 and AHEI-2010); and all-cause mortality (AHEI-2010 only). We also observed an inverse association between DQIs and CHD only in univariate analysis.

In the current study, a higher AHEI-2010 score was inversely associated with DM (from S5 to S8), with a lower odd ratio than either HEIFA-2013 or MDS. The inverse association between AHEI-2010 and DM is consistent with previous findings among middle-aged women [27,61] and postmenopausal women [62]. One reason for the stronger inverse association may be the unique components of the AHEI-2010 such as SSBs, red and processed meats, trans-fat, PUFA, and long chain omega-3 fats. Findings from previous studies suggested the increased risk of DM associated with SSBs [63], red and processed meats [64], and trans-fat [65], whereas there is no clear evidence of the preventive role of PUFA and long chain omega-3 fats intake on DM [66].

We did not find any association between DQIs and CHD in middle-aged women, after multivariate adjustment (from S5 to S8). These results contrast those of the Nurses’ Health Study (NHS) [27] and the Atherosclerosis Risk in Communities (ARIC) cohort aged 45–64 years [67,68], where AHEI-2010 was inversely associated with incident CHD. In both our analysis and the European Prospective Investigation into the Cancer and Nutrition (EPIC) cohort [69], CHD incidence was not associated with the Mediterranean Diet Score (MDS). These results contrast with findings from middle-aged women in the NHS where alternative MDS was inversely associated with incident CHD (RR_Q5-Q1_: 0.71, 95% CI 0.62, 0.82) [70] and findings from the ARIC cohort showing that those who adhered to a Mediterranean diet had 16% reduced risk of incident CVD, including CHD [67]. Of interest, there is little range in the measure of MDS (scored 0–9) compared with other DQIs (only a 1-point mean difference across quintiles 2–4 in the current study), which may partially explain the lack of association between MDS and CHD.

Previous studies conducted among the middle-aged ALSWH cohort [71,72] suggested protective effects of three different DQIs: Dietary Inflammatory Index (DII) [71]; ARFS [72]; and MDS [72] for HT. The present study has shown that three additional DQIs were similarly inversely associated with HT. For example, our results showed significantly lower odds of having HT in later surveys (S6–S8) for participants who had the highest MDS quintile, compared with those having the lowest MDS quintile, suggesting possible long-term effect of DQ on HT. All three DQIs emphasise healthful foods for HT such as fruits [73], vegetables [73], and legumes [74], and include beneficial foods and nutrients for blood pressure (i.e.,HEIFA-2013: MUFA and PUFA; MDS: fish and seafood, and lipid ratio; AHEI-2010: omega-3 fatty acids and PUFA) [75].

Previous studies have demonstrated that fruit and vegetables, whole grains, and PUFA are beneficial for asthma, and lowering oxidative stress, pulmonary dysfunction, airway inflammation, and allergy [76]. In the present study, we found an inverse association between AHEI-2010 and asthma. In contrast, there was no association between AHEI-2010 and adult-onset asthma among 73,228 female participants from the NHS [77], nor among 12,687 adult Latino participants [78]. However, in the French population-based NutriNet-Santè (NNS) cohort study, AHEI-2010 was inversely associated with asthma symptom score, which assessed the number of respiratory symptoms in the previous 12 months [79]. The discrepant findings could be explained by use of different assessments for dietary exposure such as FFQ [77], 24 h dietary recalls [78,79], and outcomes such as dichotomous measure [77] and continuous measure [78,79].

The DQIs investigated in our study gave points for beneficial foods for NCD such as fruit, vegetables [73,80,81,82,83,84], and legumes [74,85,86,87]. Underlying protective mechanisms for cancer may be contributed by (1) antioxidant actions of polyphenols and phytochemicals present in fruit, vegetables and whole grains; and (2) reduction or inhibition of insulin resistance, intestinal absorption of cholesterol and hepatic cholesterol synthesis by fibre present in fruit, vegetables, whole grains, and legumes [35]. No association between reported consumption of these foods or nutrients and incident cancer was evident in our study participants. By contrast, participants in the NHS scoring in the highest AHEI-2010 quintile had lower risk of overall cancer compared with their counterparts with lower scores (RR: 0.93, 95% CI: 0.88, 0.99) [27]. Similarly in the NNS cohort, a 10-point increase in the AHEI-2010 score was associated with a 5% lower risk of overall cancer [88]. These study results tend to align with the current epidemiological findings of inverse association between high DQ and cancer risk [31] and a reduced risk of overall cancer mortality/incidence among those adhering most closely to the Mediterranean diet [89].

MUFA and omega-3 fatty acids from fish intake present in the Mediterranean diet may contribute beneficially to depression pathology by playing an integral part in central nervous system membranes, methylation reactions, serotonin, and other neurotransmission, oxidative stress reduction, and anti-inflammation [90]. This has been supported in our study and other studies with follow-ups of 3 years [91], 4.4 years [92], and 12 years [93], but not 8.5 years [90]. However, the Mediterranean style pattern from factor analysis did not show any association with psychological distress measured by the Kessler Psychological Distress Scale (K10) [93]. The conflicting findings between studies could be explained by different methodological approaches to defining the Mediterranean diet, i.e., priori score ranging 0–9 [90,92,93], factor analysis [91,93].

There are few studies investigating the association between dietary factors and multimorbidity [94,95,96,97,98], particularly in middle-aged women. Shi et al. (2014) reported that the relative odds of multimorbidity were increased for those who consumed soft drink > 0.5 litre per day compared with those who did not consume soft drinks (OR: 2.03; 95% CI: 1.68, 2.45) [94]. Protective effects of fruit and vegetable consumption on multimorbidity have been observed in cross-sectional [95,96] and prospective [97,98,99] studies. Among European participants, the relative odds of multimorbidity decreased for those who adhered to modified MDS compared with who did not [40,100]. We observed lower odds of multimorbidity among those with higher DQ assessed by HEIFA-2013 (at S8) and AHEI-2010 (at S7, S8), but not with MDS.

An inverse association between AHEI (including AHEI-2010) and all-cause mortality has been established [31,67,101]. In our cohort, women in the highest category of AHEI-2010 had a lower risk of all-cause mortality at S4 in univariate analysis, but this disappeared when covariates were controlled for. Cumulative evidence from 29 cohort studies reported the relationship between Mediterranean diet adherence and reduction in all-cause mortality [38]; however, there was still conflicting evidence among studies using the scoring system proposed by Trichopoulou et al., score ranging 0–9 [53]. Inverse association between MDS and mortality have been reported in Dutch women [102], in the EPIC cohort [103], and an Australian cohort [104]. By contrast, no association between MDS and all-cause mortality was observed in US participants [105], Swiss women [106], and Caucasian participants [107]. It may be that the MDS does not adequately differentiate DQ in populations that do not regularly follow a Mediterranean dietary pattern [108].

Analysis of the overall diet through three DQIs with different descriptions, compositions and construction criteria based on our recent systematic review and critical appraisal [48] is a strength of the current study. DQIs can evaluate the quality and/or variety of the overall diet and the agreement with national dietary guidelines or specific dietary patterns, and importantly, they can be used to explore diet–health relationships [14]. In the present study, AHEI-2010, the most up-to-date version based on the current scientific evidence, was most frequently associated with reduced odds of having more NCDs. Another strength is that the study was based on a prospective study of nationally representative middle-aged Australian women, who were free of NCD before the start of the study.

There are some limitations that need to be considered. First, compared with originally recruited women at S1 (*n* = 13,714), women who were excluded from the analysis due to incomplete FFQ at S3 (*n* = 593, approximately 11% of S3 sample) were more likely to have no formal education, no paid job, difficulty managing on income, be less physically active, have poor self-rated health, and take OTC medicine (Appendix A). This could affect the representativeness of middle-aged women and generalisability of this study. Excluding women who were the least healthy and had poor diet could reduce the ability to observe association, likely biasing towards the null. Second, the measurement of diet was only conducted at S3 and S7; therefore, we assumed DQIs were constant during the 15-year follow-up. DQ between these two surveys has been shown to be relatively consistent for this cohort [47]; therefore, we believe this is a reasonable assumption. Third, the NCD outcomes in our study relied on self-reported data that may be biased; however, their reliability and validity against administrative data has been reported [54,109]. Fourth, the low number of cases for our outcomes at each survey, particularly in all-cause mortality, might lead to increased uncertainty in our estimates obtained from regression models. Lastly, despite the use of DAGs to inform covariate selection for the multivariate analysis, residual confounding is still possible. Given the repeated cross-sectional study design the covariates were reported at the same surveys as the outcome measures; however, some of these measures (e.g., medication use) may reflect changes that occurred after the NCD diagnosis.

## 5. Conclusions

Higher DQ was inversely associated with incident NCD outcomes within a sample of middle-aged Australian women. The different theoretical approaches reflected in three DQIs, i.e., ADG-2013 for HEIFA-2013, and specific dietary patterns for MDS and AHEI-2010, may have different implications. Our findings suggested that DQ is an important predictor of some NCDs and applicable in NCD prevention: HEIFA-2013 for HT, depression and/or anxiety, and multimorbidity; MDS for HT and depression and/or anxiety; and AHEI-2010 for DM, HT, asthma, and multimorbidity. Specific dietary patterns captured by different DQI may be useful in setting behaviour change goals for the prevention of NCDs and multimorbidity.

## Figures and Tables

**Figure 1 ijerph-18-11375-f001:**
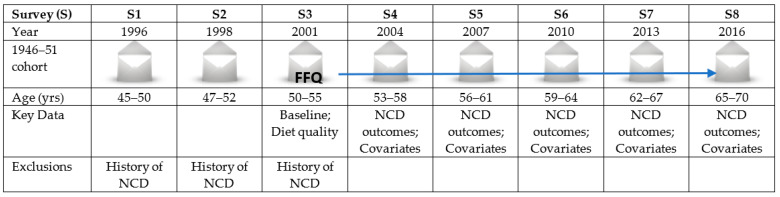
Study overview: data from the Australian Longitudinal Study on Women’s Health (ALSWH), born between 1946 and 1951.

**Figure 2 ijerph-18-11375-f002:**
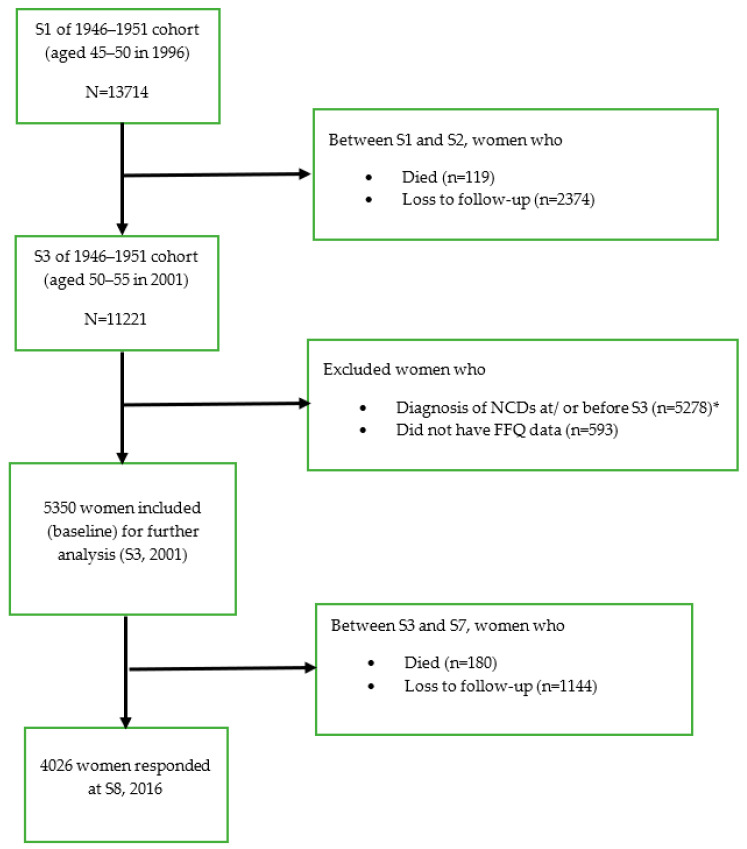
Flow diagram showing the selection of participants from the Australian Longitudinal Study on Women’s Health (ALSWH), born between 1946 and 1951. * NCDs at or before S3 were diabetes mellitus, coronary heart disease, hypertension, asthma, and cancer (excluding skin cancer).

**Table 1 ijerph-18-11375-t001:** Sociodemographic and lifestyle characteristics at study baseline (Survey 3 in 2001) related to the first (Q1) and fifth (Q5) quintiles of diet quality indices of the sampled women (*n* = 5350).

	HEIFA-2013			MDS			AHEI-2010		
Characteristics *	Q1 (*n* = 1056)	Q5 (*n* = 1059)	*p*-Value	Q1 (*n* = 1769)	Q5 (*n* = 642)	*p*-Value	Q1 (*n* = 1059)	Q5 (*n* = 1139)	*p*-Value
Mean age in years (sd)	52.38 (1.43)	52.58 (1.45)	0.004 ******	52.42 (1.45)	52.58 (1.44)	0.15	52.34 (1.45)	52.53 (1.43)	0.004 **
Marital status			0.072			0.47			0.13
Never married	30 (2.8)	33 (3.1)		49 (2.8)	25 (3.9)		29 (2.7)	44 (3.9)	
Married/de facto	865 (82.0)	866 (81.9)		1451 (82.2)	536 (83.8)		897 (84.7)	918 (80.9)	
Separated/divorced/widowed	160 (15.2)	159 (15.0)		266 (15.0)	79 (12.3)		133 (12.6)	173 (15.2)	
Area of residence			0.070			0.063			0.003 **
Urban	389 (37.0)	334 (31.7)		576 (32.7)	235 (36.6)		334 (31.7)	446 (39.3)	
Inner regional	440 (41.9)	438 (41.5)		739 (42.0)	257 (40.0)		451 (42.7)	438 (38.6)	
Outer regional/rural	222 (21.1)	283 (26.8)		446 (25.3)	150 (23.4)		270 (25.6)	252 (22.1)	
Education			<0.001^**^			<0.001 **			<0.001 **
No formal education	168 (15.9)	120 (11.3)		267 (15.1)	61 (9.5)		181 (17.1)	106 (9.2)	
High school certificate	489 (46.4)	454 (42.9)		899 (50.9)	235 (36.6)		525 (49.7)	441 (38.8)	
Apprenticeship/diploma)	201 (19.1)	251 (23.7)		342 (19.4)	155 (24.1)		217 (20.5)	275 (24.2)	
University/higher degree	196 (18.6)	233 (22.1)		257 (14.6)	191 (29.8)		134 (12.7)	316 (27.8)	
Occupation			0.032 **			0.14			<0.001 **
No paid job	263 (26.1)	207 (20.4)		407 (24.1)	129 (20.7)		275 (27.1)	203 (18.5)	
Paid job	744 (73.9)	808 (79.6)		1283 (75.9)	495 (79.3)		739 (72.9)	892 (81.5)	
Ability to manage income			<0.001 **			<0.001 **			<0.001 **
Easy/not bad	644 (61.3)	744 (70.5)		1082 (61.7)	447 (70.0)		622 (59.4)	806 (71.1)	
Sometimes/always difficult	406 (38.7)	311 (29.5)		672 (38.3)	192 (30.0)		425 (40.6)	327 (28.9)	
Physical activity			<0.001 **			<0.001 **			<0.001 **
Nil/sedentary	205 (20.1)	101 (9.8)		344 (20.2)	48 (7.6)		228 (22.4)	85 (7.7)	
Low	362 (35.6)	297 (28.8)		576 (33.9)	165 (26.3)		367 (36.1)	318 (28.8)	
Moderate	191 (18.8)	255 (24.7)		345 (20.3)	152 (24.2)		187 (18.4)	284 (25.7)	
High	260 (25.5)	379 (36.7)		436 (25.6)	263 (41.9)		236 (23.2)	419 (37.8)	
Smoking status			<0.001 **			<0.001 **			<0.001 **
Never smoked	561 (53.2)	720 (68.4)		1046 (59.2)	405 (63.5)		618 (58.4)	722 (63.8)	
History of smoking	256 (24.2)	246 (23.4)		362 (20.5)	177 (27.7)		209 (19.8)	324 (28.6)	
Currently smoke	239 (22.6)	87 (8.2)		360 (20.3)	56 (8.8)		231 (21.8)	86 (7.6)	
Taking prescribed medicine			0.92			0.56			0.005 **
Not taken	582 (55.5)	597 (56.5)		955 (54.3)	357 (55.9)		586 (55.8)	663 (58.6)	
Taken	467 (44.5)	459 (43.5)		803 (45.7)	282 (44.1)		465 (44.2)	469 (41.4)	
Taking over the counter medicine			0.12			<0.001 **			<0.001 **
Not taken	288 (27.4)	240 (22.7)		505 (28.6)	133 (20.8)		326 (30.9)	248 (21.9)	
Taken	763 (72.6)	815 (77.3)		1258 (71.4)	506 (79.2)		728 (69.1)	887 (78.1)	

* Due to missing data, the sum for each characteristic may not equal *n*. Participant characteristics vary over time, whereas diet is held constant. ** Statistically significant (*p* < 0.05). Values for categorical variables are given as “number (percentage); *n* (%)”; for continuous variable as “mean (standard deviation): mean (sd)”.

**Table 2 ijerph-18-11375-t002:** Associations between diet quality indices and risk of common NCDs (including multimorbidity) among 1946–1951 ALSWH cohort women (from S4 to S8).

Diet Quality Index	S4 (*n* = 4347) ^b^	S5 (*n* = 4168) ^b^	S6 (*n* = 4015) ^b^	S7 (*n* = 3948) ^b^	S8 (*n* = 3905) ^b^
Number of missing values	519	598	532	306	121
	OR (95% CI)	OR (95% CI)	OR (95% CI)	OR (95% CI)	OR (95% CI)
**DM**	** *n = 42* **	** *n = 119* **	** *n = 206* **	** *n = 281* **	** *n = 375* **
HEIFA-2013: Univariate	0.54 (0.18–1.63)	0.77 (0.42–1.39)	0.93 (0.59–1.46)	0.86 (0.58–1.27)	0.70 (0.49–1.00)
: Multivariate	0.71 (0.20–2.51)	1.03 (0.52–2.05)	1.06 (0.63–1.77)	0.95 (0.62–1.43)	0.76 (0.52–1.10)
MDS: Univariate	1.33 (0.40–4.44)	0.62 (0.28–1.34)	0.96 (0.57–1.62)	0.74 (0.46–1.21)	0.60 (0.39–0.94) *
: Multivariate	1.70 (0.42–6.88)	0.94 (0.42–2.11)	1.30 (0.73–2.31)	0.95 (0.57–1.60)	0.76 (0.48–1.21)
AHEI-2010: Univariate	0.71 (0.26–1.92)	0.34 (0.18–0.66) *	0.44 (0.27–0.73) *	0.43 (0.28–0.67) *	0.35 (0.23–0.51) *
: Multivariate	1.00 (0.30–3.29)	0.50 (0.25–0.99) *	0.58 (0.33–0.99) *	0.51 (0.31–0.84) *	0.44 (0.29–0.66) *
**CHD**	** *n = 58* **	** *n = 136* **	** *n = 214* **	** *n = 320* **	** *n = 409* **
HEIFA-2013: Univariate	1.21 (0.58–2.53)	1.12 (0.65–1.93)	0.63 (0.40–0.97) *	0.92 (0.63–1.33)	0.92 (0.65–1.29)
: Multivariate	1.34 (0.60–3.00)	1.11 (0.62–1.98)	0.72 (0.44–1.18)	1.21 (0.81–1.82)	1.01 (0.70–1.46)
MDS: Univariate	0.94 (0.37–2.39)	0.48 (0.24–0.99) *	0.53 (0.31–0.91) *	0.77 (0.50–1.18)	0.79 (0.54–1.15)
: Multivariate	0.88 (0.28–2.73)	0.58 (0.28–1.20)	0.68 (0.38–1.22)	0.99 (0.63–1.56)	0.94 (0.64–1.39)
AHEI-2010: Univariate	1.12 (0.46–2.72)	0.67 (0.38–1.18)	0.50 (0.32–0.81) *	0.58 (0.40–0.84) *	0.65 (0.46–0.91) *
: Multivariate	1.24 (0.48–3.25)	0.72 (0.39–1.33)	0.63 (0.38–1.06)	0.74 (0.50–1.11)	0.78 (0.54–1.12)
**HT**	** *n = 244* **	** *n = 604* **	** *n = 924* **	** *n = 1194* **	** *n = 1419* **
HEIFA-2013: Univariate	0.83 (0.55–1.24)	0.69 (0.53–0.90) *	0.76 (0.60–0.96) *	0.75 (0.60–0.93) *	0.71 (0.57–0.88) *
: Multivariate	0.93 (0.59–1.44)	0.77 (0.57–1.04)	0.83 (0.64–1.09)	0.74 (0.58–0.94) *	0.65 (0.51–0.82) *
MDS: Univariate	1.01 (0.66–1.56)	0.86 (0.64–1.17)	0.65 (0.50–0.85) *	0.68 (0.53–0.87) *	0.73 (0.58–0.91) *
: Multivariate	1.23 (0.77–1.97)	1.19 (0.85–1.66)	0.73 (0.55–0.98) *	0.75 (0.57–0.98) *	0.76 (0.59–0.97) *
AHEI-2010: Univariate	0.54 (0.36–0.81) *	0.64 (0.49–0.85) *	0.65 (0.51–0.82) *	0.69 (0.56–0.86) *	0.66 (0.54–0.82) *
: Multivariate	0.67 (0.43–1.04)	0.77 (0.57–1.05)	0.82 (0.63–1.07)	0.79 (0.62–1.01)	0.74 (0.59–0.94) *
**Asthma**	** *n = 76* **	** *n = 159* **	** *n = 243* **	** *n = 314* **	** *n = 374* **
HEIFA-2013: Univariate	0.73 (0.37–1.44)	0.99 (0.62–1.59)	1.18 (0.78–1.78)	0.97 (0.67–1.40)	0.98 (0.69–1.39)
: Multivariate	1.03 (0.49–2.17)	1.05 (0.63–1.77)	1.09 (0.69–1.71)	1.14 (0.77–1.71)	1.09 (0.75–1.57)
MDS: Univariate	0.63 (0.26–1.56)	1.20 (0.68–2.11)	0.92 (0.57–1.47)	0.82 (0.53–1.26)	0.89 (0.60–1.32)
: Multivariate	0.93 (0.37–2.35)	1.49 (0.81–2.73)	1.11 (0.67–1.83)	0.99 (0.63–1.57)	1.05 (0.70–1.58)
AHEI-2010: Univariate	0.56 (0.28–1.13)	0.76 (0.46–1.25)	0.68 (0.45–1.03)	0.56 (0.38–0.83) *	0.56 (0.38–0.81) *
: Multivariate	0.84 (0.39–1.80)	0.76 (0.44–1.31)	0.72 (0.46–1.13)	0.65 (0.43–0.99) *	0.63 (0.43–0.93) *
**Cancer (excludes skin cancer)**	** *n = 86* **	** *n = 199* **	** *n = 304* **	** *n = 417* **	** *n = 555* **
HEIFA-2013: Univariate	0.87 (0.44–1.72)	1.44 (0.91–2.27)	1.31 (0.89–1.91)	1.31 (0.93–1.83)	1.18 (0.87–1.60)
: Multivariate	0.94 (0.46–1.95)	1.52 (0.90–2.56)	1.25 (0.82–1.90)	1.40 (0.97–2.00)	1.25 (0.91–1.72)
MDS: Univariate	0.62 (0.27–1.41)	0.83 (0.49–1.40)	0.88 (0.57–1.36)	1.06 (0.73–1.52)	1.03 (0.75–1.42)
: Multivariate	0.58 (0.24–1.43)	0.85 (0.48–1.51)	0.90 (0.56–1.43)	1.13 (0.77–1.66)	1.10 (0.79–1.53)
AHEI-2010: Univariate	1.64 (0.82–3.25)	1.20 (0.76–1.91)	1.02 (0.69–1.51)	1.11 (0.78–1.59)	1.21 (0.88–1.65)
: Multivariate	1.62 (0.78–3.36)	1.30 (0.78–2.18)	0.94 (0.61–1.46)	1.19 (0.81–1.74)	1.28 (0.93–1.77)
**Depression/anxiety**	** *n = 622* **	** *n = 670* **	** *n = 644* **	** *n = 602* **	** *n = 559* **
HEIFA-2013: Univariate	0.78 (0.60–1.03)	0.87 (0.67–1.13)	0.79 (0.60–1.03)	0.80 (0.61–1.04)	0.77 (0.58–1.03)
: Multivariate ^a^	0.96 (0.69–1.35)	1.03 (0.74–1.41)	0.70 (0.51–0.98) *	0.81 (0.58–1.12)	0.78 (0.55–1.10)
MDS: Univariate	0.91 (0.68–1.21)	1.01 (0.77–1.32)	0.87 (0.65–1.15)	0.66 (0.49–0.90) *	0.72 (0.53–0.97) *
: Multivariate ^a^	1.03 (0.72–1.47)	1.09 (0.78–1.52)	0.84 (0.59–1.18)	0.70 (0.49–0.96) *	0.66 (0.46–0.94) *
AHEI-2010: Univariate	0.95 (0.72–1.26)	1.05 (0.80–1.37)	0.94 (0.71–1.25)	0.99 (0.74–1.32)	0.93 (0.69–1.25)
: Multivariate ^a^	1.12 (0.80–1.58)	1.09 (0.78–1.51)	0.95 (0.67–1.35)	1.07 (0.76–1.50)	1.07 (0.75–1.53)
**Multimorbidity**	** *n = 133* **	** *n = 300* **	** *n = 473* **	** *n = 657* **	** *n = 857* **
HEIFA-2013: Univariate	0.65 (0.37–1.12)	0.65 (0.44–0.94) *	0.76 (0.56–1.04)	0.77 (0.59–1.01)	0.70 (0.54–0.90) *
: Multivariate ^a^	0.86 (0.47–1.59)	0.70 (0.46–1.08)	0.78 (0.54–1.10)	0.90 (0.66–1.21)	0.73 (0.55–0.96) *
MDS: Univariate	1.13 (0.60–2.13)	0.80 (0.51–1.26)	0.64 (0.44–0.92) *	0.66 (0.48–0.90) *	0.75 (0.57–0.98) *
: Multivariate ^a^	1.25 (0.60–2.59)	1.09 (0.66–1.80)	0.76 (0.51–1.14)	0.83 (0.58–1.17)	0.87 (0.65–1.17)
AHEI-2010: Univariate	0.72 (0.41–1.26)	0.68 (0.46–1.00)	0.74 (0.54–1.02)	0.56 (0.43–0.74) *	0.63 (0.49–0.81) *
: Multivariate ^a^	0.93 (0.50–1.74)	0.73 (0.47–1.12)	0.89 (0.63–1.26)	0.70 (0.51–0.96) *	0.75 (0.57–0.99) *

CHD: coronary heart disease; CI: confidence interval; DM: diabetes mellitus; HT: hypertension; NCD: non-communicable disease; OR: odds ratio. OR (95% CI) described in the table is the odds of having NCDs (each disease, multimorbidity) compared with quintile 5 to quintile 1 of DQIs (HEIFA-2013, MDS and AHEI-2010). Adjusted covariates were age; socioeconomic status (marital status, residence, education, occupation, and ability to manage income); lifestyle variables (smoking status, physical activity, taking prescribed and over-the-counter medicine) for all NCD outcomes. ^a^ History of depression and/or anxiety at any previous survey(s) was included as a covariate; ^b^ Number in parenthesis is number of women in each survey for multivariate analysis; Number in bold and italic are cumulative number of NCD cases (except depression/ anxiety) in each survey; * Statistically significant (*p* < 0.05).

**Table 3 ijerph-18-11375-t003:** Associations between diet quality indices and risk of all-cause mortality among 1946–51 ALSWH cohort women (from S4 to S8).

Diet Quality Index	OR (95% CI)	OR (95% CI)	OR (95% CI)	OR (95% CI)	OR (95% CI)
All-Cause Mortality in all Women					
** *Number of deaths at each survey* **	** *n = 32* **	** *n = 28* **	** *n = 49* **	** *n = 55* **	** *n = 31* **
**HEIFA-2013**	S4 (*n* = 5334) ^a^	S5 (*n* = 5302) ^a^	S6 (*n* = 5274) ^a^	S7 (*n*= 5225) ^a^	S8 (*n* = 5170) ^a^
Univariate	0.98 (0.37–2.61)	0.73 (0.30–1.79)	0.93 (0.45–1.94)	0.54 (0.28–1.07)	0.37 (0.14–1.01)
	S4 (*n* = 3618) ^b^	S5 (*n* = 4180) ^b^	S6 (*n* = 4024) ^b^	S7 (*n*= 3955) ^b^	S8 (*n* = 3906) ^b^
Multivariate *	2.03 (0.53–7.77)	0.69 (0.21–2.24)	0.72 (0.27–1.93)	0.63 (0.26–1.50)	0.44 (0.13–1.48)
**MDS**	S4 (*n* = 5334) ^a^	S5 (*n* = 5302) ^a^	S6 (*n* = 5274) ^a^	S7 (*n*= 5225) ^a^	S8 (*n* = 5170) ^a^
Univariate	0.57 (0.20–1.64)	0.57 (0.22–1.52)	0.45 (0.20–1.01)	0.80 (0.37–1.71)	0.38 (0.14–1.05)
	S4 (*n* = 3618) ^b^	S5 (*n* = 4180) ^b^	S6 (*n* = 3240) ^b^	S7 (*n*= 3955) ^b^	S8 (*n* = 3906) ^b^
Multivariate *	1.64 (0.48–5.67)	0.33 (0.07–1.57)	0.42 (0.16–1.10)	0.92 (0.32–2.63)	0.61 (0.18–2.01)
**AHEI-2010**	S4 (*n* = 5334) ^a^	S5 (*n* = 5302) ^a^	S6 (*n* = 5274) ^a^	S7 (*n*= 5225) ^a^	S8 (*n* = 5170) ^a^
Univariate	0.26 (0.09–0.78) **	0.94 (0.32–2.76)	0.98 (0.48–2.02)	0.62 (0.32–1.22)	0.51 (0.22–1.21)
	S4 (*n* = 3618) ^b^	S5 (*n* = 4180) ^b^	S6 (*n* = 4024) ^b^	S7 (*n*= 3955) ^b^	S8 (*n* = 3906) ^b^
Multivariate *	0.88 (0.23–3.41)	0.55 (0.12–2.61)	0.89 (0.34–2.36)	1.43 (0.52–3.88)	1.14 (0.37–3.46)
**All-cause Mortality in women with NCD**					
**HEIFA-2013**	S4 (*n* = 1241) ^c^	S5 (*n* = 1709) ^c^	S6 (*n* = 2085) ^c^	S7 (*n*= 2414) ^c^	S8 (*n* = 2660) ^c^
Univariate	0.86 (0.20–3.63)	0.53 (0.15–1.84)	0.83 (0.32–2.16)	0.67 (0.30–1.49)	0.38 (0.13–1.10)
	S4 (*n* = 573) ^d^	S5 (*n* = 827) ^d^	S6 (*n* = 1394) ^d^	S7 (*n* = 1923) ^d^	S8 (*n* = 2139) ^d^
Multivariate	1.02 (0.16–6.54)	0.19 (0.03–1.05)	0.63 (0.19–2.15)	0.76 (0.29–1.95)	0.52 (0.15–1.80)
**MDS**	S4 (*n* = 1241) ^c^	S5 (*n* = 1709) ^c^	S6 (*n* = 2085) ^c^	S7 (*n* = 2414) ^c^	S8 (*n* = 2660) ^c^
Univariate	0.41 (0.04–3.99)	0.74 (0.21–2.53)	0.41 (0.13–1.26)	0.87 (0.31–2.44)	0.50 (0.18–1.42)
	S4 (*n* = 713) ^d^	S5 (*n* = 827) ^d^	S6 (*n* = 1131) ^d^	S7 (*n* = 1923) ^d^	S8 (*n* = 2139) ^d^
Multivariate	0.93 (0.07–11.88)	0.21 (0.02–1.94)	0.35 (0.09–1.29)	0.96 (0.29–3.12)	0.68 (0.20–2.29)
**AHEI-2010**	S4 (*n* = 1241) ^c^	S5 (*n* = 1709) ^c^	S6 (*n* = 2085) ^c^	S7 (*n*= 2414) ^c^	S8 (*n* = 2660) ^c^
Univariate	0.32 (0.05–1.92)	0.70 (0.16–3.16)	0.66 (0.26–1.69)	1.64 (0.60–4.49)	0.46 (0.18–1.20)
	S4 (*n* = 713) ^d^	S5 (*n* = 827) ^d^	S6 (*n* = 1394) ^d^	S7 (*n* = 1923) ^d^	S8 (*n* = 2139) ^d^
Multivariate	0.97 (0.08–11.75)	0.17 (0.02–1.80)	0.54 (0.16–1.80)	3.61 (0.79–16.54)	1.05 (0.34–3.28)

^a^ Number in parenthesis is number of all women in each survey minus deaths in preceding survey for univariate analysis; ^b^ Number in parenthesis is number of all women in each survey minus deaths in preceding survey for multivariate analysis; ^c^ Number in parenthesis is number of women with NCD in each survey for univariate analysis; ^d^ Number in parenthesis is number of women with NCD in each survey for multivariate analysis; Number in bold and italic are number of all-cause mortality in each survey; OR: odds ratio. OR (95% CI) described in the table is the odds of all-cause mortality comparing the highest category to the lowest category of DQIs (HEIFA-2013, MDS and AHEI-2010). Quintiles 4 and 5 were collapsed due to their small sample sizes. Adjusted covariates were age; socioeconomic status (residence, education, occupation and ability to manage income); lifestyle variables (smoking status, physical activity, taking prescribed and over-the-counter medicine), and history of depression and/or anxiety at any previous survey(s). * Presence of NCD at each survey was included as a covariate; ** Statistically significant (*p* < 0.05).

## Data Availability

The information about data access to the Australian Longitudinal Study on Women’s Health is available at https://www.alswh.org.au/ [43].

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
