# Peer review of "Diet Quality and Incident Non-Communicable Disease in the 1946–1951 Cohort of the Australian Longitudinal Study on Women’s Health"

_ijerph, 2021, doi:10.3390/ijerph182111375_

Round 1

Reviewer 1 Report

This study presents exciting research on an important topic. Very well written. It would contribute to the literature on diet quality of health outcomes. I have a few concerns and comments. Please see below. 

 The authors do not provide sufficient background information on the importance of diet quality measuring indices concerning health outcomes. Since diet quality itself does not affect non-communicable diseases, authors could emphasize the mechanism through which they are interconnected. For example, previous studies have shown that diet quality, such as Healthy Eating Index, is strong and positively associated with obesity and higher BMI, which is a major risk factor of several non-communicable diseases. That would make your story more interesting in the introduction. 

Also, it is well known that individuals with low socioeconomic status (SES) are more likely to eat a poor diet and develop non-communicable diseases. Although outside the scope of this study, you can give a little bit of background info on the differential effects across SES. 

A substantial body of literature has established these relationships. For example, you can see the following recent literature:

Dhakal, C.K.; Khadka, S. Heterogeneities in Consumer Diet Quality and Health Outcomes of Consumers by Store Choice and Income. Nutrients 202113, 1046. https://doi.org/10.3390/nu13041046

Aggarwal, A., Monsivais, P., Cook, A. J., & Drewnowski, A. (2011). Does diet cost mediate the relation between socioeconomic position and diet quality?. European journal of clinical nutrition65(9), 1059-1066.

There are several typos throughout the text—For example, extra space after "included .." in line 16. 

line 29: Please replace "confirm" with another word or phrase.-maybe "support evidence that" 

You don't have to report missing values in Table 2. Also, maintain consistency in column 1 of Table 2 as in the AHEI-2010. 

Why are there empty rows in Table 3? One concern is the power of the test given the small sample size in each column of Table 3. 

I am not sure what each column of the Table 3 represents. Also, there are several empty rows.  

Please make sure the text inside figure 2 is consistent with the text. 

The discussion is too long. Please make it short and concise. 

Would you please check the reference style and make sure it follows the journal's guidelines? 

Reviewer 2 Report

Thank you for preparing this article on diet quality and incident of non-communicable disease in representative sample of Australian women.

It is important to gain an understanding of the impact diet has on disease, papers like this highlight the importance of everyday activies on long term health.  Clearly alot of work has gone into this ongoing study and the preparation of the paper.  Well done

I have a few comments for your consideration:

Abstract:

Given you are testing the theory of whether or not DQI influence NCD you should change the word "are" useful to "can be" useful. 

Line 29: ad the word "some" infont of NCDs

Introduction:

The flow is lost between paragraph 2 and 3.  It would be good to have some explanation around how HEI and AHEI fit into the country developed DQI mentioned in paragraph 2 and why these have perhaps been used instead of country specific indices.

Materials and Methods:

It would help with flow and understanding to put the sample number used in the current study somewhere in or after the sentence that starts on line 90 and ends on line 92.

You should include a reference afte DQES-v2 when first mentioned on line 97.

After physical activity is mentioned in line 316 it would be good to include exactly the questionnaire used to measure physical activity. I can see the questions asked from the provided link - I am assuming this is similar to the short IPAQ but was it a specific questionnaire or just questions? The reference you have given assess 4 different types. Please clarify.

Results:

Table 1: 

Please label or title the table appropriately so that it can stand alone. on the table in thelengend or in the title should say that the numbers represent n and the brackets represent percentage. Did you do post hoc testing to tell where the differences were between the multiply options within each category? Adding this to the table would also make things clearer.

Discussion:

The discussion generally would benefit from being a bit more concise and to have a clearer picture of what the three different indices tell collectively and why there are differences.

Perhaps something your study does suggest is using DQI that are closer to the countries dietary guidelines gives better predictive value. As per Line 565 you have mentioned that MDS may not adequately differentiate in populations that do not regularly follow a Mediterranean dietary pattern. Or more clearly live outside the Mediterranean region. Suggesting that there may be other factors in their lifestyle that are predictive/protective.

You may like to add a comment about the length of time it takes see significant reduction in risk and thus the importance of maintaining healthful lifestyle habits.

Line 547 after the word analysis consider adding: but this disappeared when covariants were controlled for.

Line 548 between the word was and inverse add the word “an”.

In the limitation section: line 592 consider : removing the word however, and add to the end of the sentence line 593 “therefore we believe this is a reasonable assumption”.

Round 2

Reviewer 1 Report

Authors have addressed all my concerns/comments. No further comments. It warrants publication. Thanks.